# Excessive exercise induces cardiac arrhythmia in a young fibromyalgia mouse model

**Taiki Nakata[1,2]⊘, Atsushi Doi [iD][2,3]⊘\*, Daisuke Uta[4], Megumu Yoshimura[5], Min-Chul Shin[2,3]**

**1** Department of Rehabilitation, Kumamoto-Saiseikai Hospital, Kumamoto, Japan, **2** Graduate school of Health Science, Kumamoto Health Science University, Kumamoto, Japan, **3** Department of Rehabilitation, Kumamoto Health Science University, Kumamoto, Japan, **4** Department of Applied Pharmacology, Faculty of Pharmaceutical Sciences, University of Toyama, Sugitani, Toyama, **5** Department of Orthopedic Surgery, Nakamura Hospital, Fukuoka, Japan

⊘ These authors contributed equally to this work.
\* atsushidoi01@gmail.com

## Abstract

### Background

Fibromyalgia patients experience cardiovascular complications in addition to musculoskeletal pain. This study aimed to investigate the cardiac effects of a prolonged shallow water gait in a fibromyalgia-induced young mouse model.

### Methods

To produce a fibromyalgia mouse model, wild-type mice were administered an intraperitoneal injection of reserpine once a day for three days, and two primary experiments were performed. First, three types of gait tests were performed before and after the reserpine injections as follows: (i) 5 minutes of free gait outside the water, (ii) 1 minute of free gait in shallow warm water, and (iii) 5 minutes of free gait in shallow warm water. Second, electrocardiogram recordings were taken before and after the three gait tests. The average heart rate and heart rate irregularity scores were analyzed.

### Results

Exercise-induced cardiac arrhythmia was observed at 1-minute gait in shallow water during the acute stage of induced FM in young mice. Further, both cardiac arrhythmia and a decrease in HR have occurred at 5-minute gait in shallow water at the same mice. However, this phenomenon was not observed in the wild-type mice under any test conditions.

### Conclusion

Although a short-term free gait in shallow warm water may be advantageous for increasing the motor activity of FM-model mice, we should be aware of the risk of prolonged and excessive exercise-induced cardiac arrhythmia. For gait exercises in shallow water as a treatment in FM patients. We suggest a gradual increase in exercise duration may be warranted.

**Data Availability Statement:** All relevant data are within the manuscript.

**Funding:** Kumamoto Health Science University fellowship grant (No. 2018-C-11) and a Grant-in-Aid for Scientific Research (C), JSPS to A.D. (19K11383).

**Competing interests:** The authors have declared that no competing interests exist.

## Introduction

Fibromyalgia (FM) is an immune system-related intractable disease with higher morbidity rates in women compared with men [1, 2]. The symptoms of FM comprise musculoskeletal pain [3], fatigue [4], sleep disorders, and mood disorders [4]. In general, medications [5, 6], psychotherapy [7], counseling [8], cognitive behavior therapy [9], self-care instructions [10, 11], and rehabilitation or therapeutic exercises [12, 13] are recommended as treatments for FM patients. Underwater walking is a therapeutic exercise that has often been utilized for gait training for patients with painful diseases, such as osteoarthritis [14], rheumatoid arthritis [15], *arthroplasty [16],* and FM [17–20]. Although the functional activity of antigravity muscles, such as the erector spinae, gluteus maximus, quadriceps, and triceps surae, is required for walking, underwater walking appears to decrease overload on the antigravity muscles.

To investigate effective treatments for FM patients, FM-induced animal models have been made for basic animal research. FM was initially induced by acidic saline injection [21], hydrochloric acid injection [22], or intermittent cold stress [23]. However, these animal models showed comparative recovery within two weeks. Reserpine, an anti-hypertensive medication, has recently been utilized for generating a FM animal model, and it appears to be effective in producing long-term symptoms of FM [24–26]. Reserpine blocks vesicular monoamine transporters. As a result, reserpine-injected FM-induced animals experience a depletion in monoamines in the presynaptic terminals of the central and peripheral synapses [25].

Recently, FM patients have been reported to have higher heart rates (HR) [27], impaired cardiac function [28], and a higher risk of coronary heart disease-related events [29]. Moreover, FM-induced rats were also observed to develop an increase in cardiac sympathetic events [30]. Thus, both FM patients and animals appear to experience cardiovascular complications. However, there have been no studies regarding gait exercise-induced cardiac arrhythmia in animal models.

Our study aimed to investigate whether a prolonged walk in shallow water can be used as a simple, therapeutic exercise in an animal model. Secondly, we aimed to test whether prolonged walking in water results in cardiac arrhythmia in a young FM-induced mouse model.

## Material and methods

### Animals

Male C57BL/6J mice (5 weeks old, 20 g, n = 39) were purchased from Kyudo, Inc. (Kumamoto, Japan) and housed at a controlled temperature (24 ± 1 ºC) and humidity (55 ± 10%) with a 12-hour light-dark cycle and freely available food and water. The Animal Care Committee of Kumamoto Health Science University approved the animal experiments. Animal experiments were conducted in accordance with the National Institute of Health's Guide for the Care and Use of Laboratory Animals (NIH publications No. 80–23, revised 1996).

### Reserpine-injected FM model mice

To induce FM in the mice, an intraperitoneal reserpine injection (1 mL/kg) was administered to wild-type mice once a day for three days [25]. Reserpine was purchased from Nacalai Tesque (No. 30013–81, Kyoto, Japan) and dissolved in 100% glacial acetic acid (1 mg/0.05 mL) (A solution). Distilled water (1 mg/0.95 mL) was subsequently added to the A solution (B solution, 1 mg/1 mL), in which 5% glacial acetic acid was contained in the B solution. Further, the B solution was diluted to a final concentration of 0.5% acetic acid with distilled water (stock solution), which was subcutaneously injected into the mice once a day at a dosage of 10 mL/kg

for three days. Therefore, one injected dose would constitute 0.3 ml of reserpine (stock solution) for a mouse with a body weight of 30 g.

## Experimental design

Three sets of experiments were performed. First, before and after the reserpine injection, the weight and rectal temperature of the FM-induced mice were recorded, and the free gait inside the cage was videotaped (weight and rectal temperature, n = 6; video recordings of free gait, n = 9). Second, three types of gait tests were performed before and after the reserpine injection (pre-reserpine injection, n = 10; post-reserpine injection, n = 6). The three gait tests were as follows: (i) 5 minutes of free gait out of the water, (ii) 1 minute of free gait in shallow warm water, and (iii) 5 minutes of free gait in shallow warm water. Third, electrocardiogram (ECG) recordings (see below "Evaluation of cardiac function with three gait tests") were obtained before and after the gait tests. In this set of experiments, ECG was performed for ten FM-induced mice because only ten mice (out of 20) were injected with reserpine. The remaining ten wild-type mice that were not injected with reserpine were used as controls. For the gait tests and the ECG recordings, the mice were first examined in the cage without water. The mice were then placed in cages with shallow water for a 1-minute gait test. After finishing the 1-minute gait test, the mice were carefully wiped with a Kim towel (60001, Nihon Paper Crecia Co, Tokyo, Japan). Two hours after finishing the 1-minute gait test, the mice performed a 5-minute gait test in shallow water. All the gait tests and ECG recordings performed before and after the generation of the FM-induced mice used the same protocol.

## Body weight

The body weight of the FM-induced mice was measured using a weighing balance (Sefi 1B-1KM, Osaka, Japan). Awake mice were allowed to walk onto the weighing balance. After the mice became stationary, the body weight was recorded from the digital display of the weighing scale once it had stabilized. Body weight was measured for up to six days after the third reserpine injection (n = 6) [31].

## Rectal temperature

Rectal temperature was used as a measure of body temperature in the mice (CTM-303, TERUMO, Tokyo, Japan). At first, awake mice were placed on top of the cage (KN-604, Natsume Seisakusyo Co, Tokyo, Japan); the roof of the cage was made of strong mesh-wire. Then, while the mice gripped the mesh-wire with both forelimbs, the examiner picked up their tails. A standard method of rectal temperature measurement, involving the insertion of a small-diameter temperature probe with Vaseline (Vaseline HG, Taiyo Seiyaku Co, Tokyo, Japan) into the anus at a depth >2 cm, was used. The temperature-sensitive probe was connected to equipment that indicated a digital temperature scale. Rectal temperatures were taken for six days after inducing FM in the model mice (n = 6) [31].

## Evaluation of locomotive behavior using three types of gait tests

Three types of gait tests were performed. The temperature of the shallow warm water was 40–42 ºC for the 1-minute and 5-minute free gait tests. Warm water was placed inside a plastic cage (KN-601-B, Natume Co. Tokyo, Japan) at a depth of 1 cm. Subsequently, movement during the three gait tests was recorded using a digital camera (100 frames/s, TZ-35, Panasonic, Osaka, Japan) placed above the cage. While tracking the gait of the mice from above, gait distance (cm/1 or 5 minutes), maximum speed (cm/s), and average speed (cm/s) were measured.

In the analysis of the gait videos, Avidemux (http://fixounet.free.fr/avidemux/), Any Video Converter (http://www.any-video-converter.com/products/for_video_free/), VirtualDub (http://www.virtualdub.org/), and ImageJ (https://fiji.sc/), which are open-source software programs, were used [37]. The videos were cut with Avidemux and edited with Any Video Converter and VirtualDub. The edited videos were then analyzed with ImageJ [37].

### Evaluation of cardiac function using three gait tests

A disposable self-adhesive Ag/AgCl snap dual-electrode (#272S, NORAXON, Scottsdale, AZ, USA) was cut into two pieces and used to detect the ECG signals before and after the gait tests. The mice were kept in the prone position by gently holding the back of their neck. While a mouse was held, one part of the electrode was placed onto the palmar surface of the right fore-foot as the positive electrode, and the other part of the electrode was placed onto the plantar surface of the left hind-foot as the negative electrode. Usually, clear ECG signals were recorded for at least 20 seconds, which suggested that 200–250 cycles on average were recorded from one mouse. The ECG signals were amplified using a differential amplifier (model 1700, AM-System, Sequim, WA, USA) and digitized using a digitizer (Axon DigiData 1322, Molecular Device, San Jose, CA, USA). The average HR (HR means numbers of "R" in a minute; see S1D Fig), standard deviation (SD) of the HR, and HR irregularity score were analyzed using DataView 11 (University of St Andrews, Scotland, UK) [32, 33]. The HR irregularity score was determined for each cycle using 50 ECG cycles with the formula (for consecutive cycle length values) stated below:

$$Sn = 100 * ABS\,(Pn - Pn - 1)/Pn - 1$$

(Sn = score of the nth cycle, Pn = period of the nth cycle, Pn-1 = period of the cycle preceding the nth cycle, ABS = the absolute value) [34, 35].

### Statistical analysis

Experimental data were expressed as average ± SD. Single comparisons were conducted using the Wilcoxon's signed-rank test and the Mann-Whitney U-test for paired and unpaired groups, respectively. Statistical significance was set at a p-value<0.05. All statistical analyses were performed using EZR (Saitama Medical Center, Jichi Medical University, Saitama, Japan), which is a graphical user interface for R (The R Foundation for Statistical Computing, Vienna, Austria). More precisely, it is a modified version of R commander designed to add statistical functions frequently used in biostatistics [36].

## Results

### The effects of reserpine injections on body condition and behavior

The three injections of reserpine resulted in a gradual decrease in the weight of the young mice (Fig 1A). However, six days after the reserpine injection, their body weight increased slightly. In addition, the rectal temperature of the FM-induced mice decreased slightly (Fig 1B). Further, regarding the free gait video analysis before and after the reserpine injections, three gait parameters, namely, distance (m) (wild-type: 4.17 ± 0.57 vs. FM: 0.53 ± 0.21, p<0.05, n = 9; Fig 1Ci), maximum speeds (cm/s) (wild-type: 33.1 ± 2.92 vs. FM: 8.94 ± 2.89, p<0.05, n = 9; Fig 1Cii) and average speeds (cm/s) (wild-type: 8.67 ± 1.20 vs. FM: 1.10 ± 0.43, p<0.05, n = 9; Fig 1Ciii), decreased after FM had been induced in the mice.

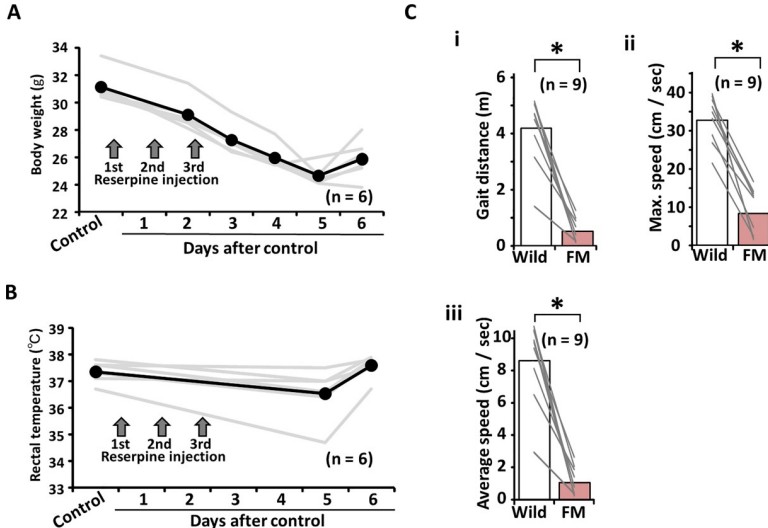

**Fig 1. The effects of reserpine injections on body condition and behavior.** (A) A line graph illustrating a change in body weight and (B) rectal temperature before and after the reserpine injection in the first set of experiments. (C) An example of gait tracking in wild-type and FM-induced mice. (Ci) Bar graphs for comparison of gait distance, (Cii) maximum speed, and (Ciii) average speed in wild-type and FM-induced model mice. FM, fibromyalgia.

## Free gait facilitation in shallow warm water in wild-type mice and FM-induced model mice

Using a video recorder, differences in the 5-minute free gait were investigated in the wild-type mice both in and out of shallow warm water (Fig 2Ai and 2Aii). Five minutes of free gait in shallow warm water was used to observe enhancement in gait distance covered (m) (out of water: 1.22 ± 0.33 vs. in shallow water: 2.06 ± 0.73, p<0.05; Fig 2Aiii), maximum speed (cm/s) (out of water: 25.46 ± 5.74 vs. in shallow water: 56.65 ± 14.47, p<0.05; Fig 2Aiv), and average speed (cm/s) (out of water: 4.57 ± 1.28 vs. in shallow water: 8.81 ± 2.86, p<0.05; Fig 2Av). Fig 1D shows that a decrease was observed in the gait distance covered and maximum speed by the same mice after the reserpine injections. However, the addition of shallow warm water into the cage improved gait distance (m) (5-minute gait out of water: 0.44 ± 0.48, 1-minute gait in shallow water: 1.43 ± 1.23, p<0.05, 5-minute gait in shallow water: 6.20 ± 0.34; Fig 2Biv) and maximum speed (cm/s) (5-minute gait out of shallow water: 3.66 ± 4.16, 1-minute gait in shallow water: 19.92 ± 4.16, p<0.05, 5-minute gait in shallow water: 31.50 ± 25.43; Fig 2Bv). Although the distance covered (m) among the FM-induced mice remained less than that covered by the wild-type mice (wild-type: 20.58 ± 7.66 vs. FM: 6.20 ± 3.42, p<0.05; Fig 2Ci), the maximum speed (cm/s) in the FM-induced model mice was not statistically different from that in the wild-type mice (wild-type: 56.65 ± 15.25 vs. FM: 31.50 ± 25.43; Fig 2Cii), even when considering the lesser maximum speed attained in the FM-induced mice.

## Cardiac effects of free gait in shallow warm water in wild-type mice

To determine the effects of the gait tests on cardiac function, ECGs were analyzed before and after the tests in the wild-type mice (Fig 3Ai and 3Aii). Under conditions of 5 minutes of free gait in and out of shallow warm water, no significant change was observed in the average HR (beats/min) (before and after the 5-minute gait out of water: 646.82 ± 18.98 and 667.38 ± 17.66; Fig 3Bi) (before and after the 5-minute gait in shallow water: 689.08 ± 20.19 and 661.95 ± 24.00; Fig 3Bii), SD of the HR (beats/min) (before and after the 5-minute gait out

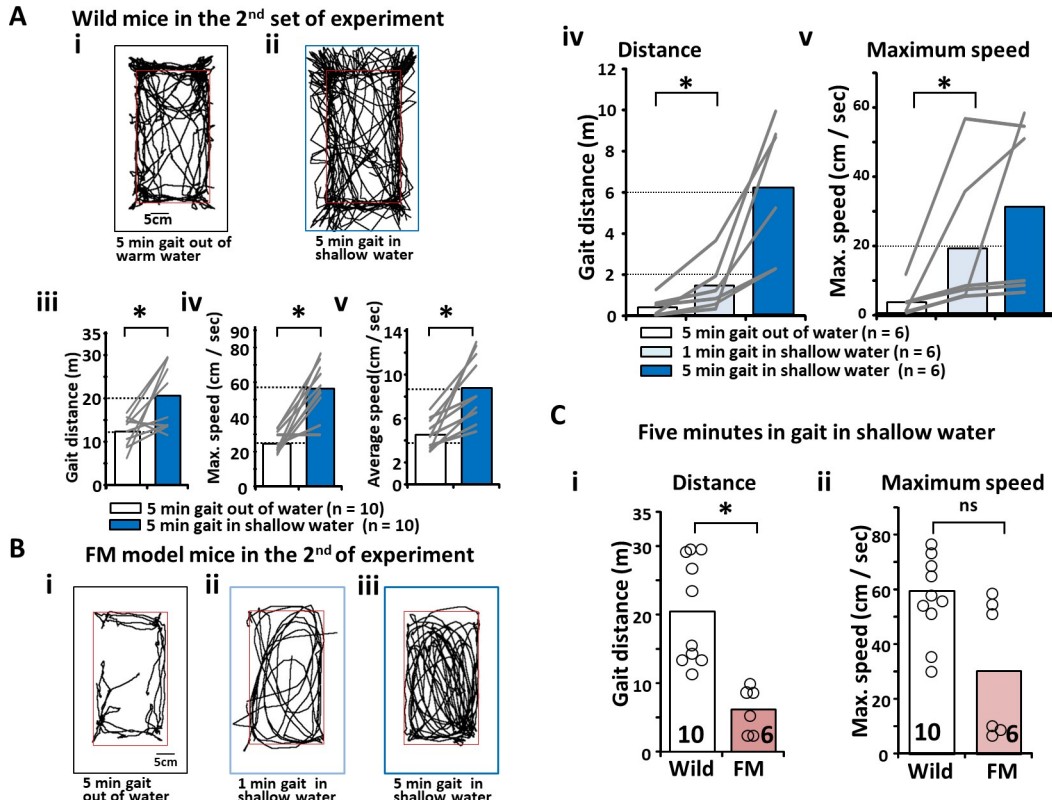

**Fig 2. Free gait facilitation in shallow warm water in wild-type mice and FM-induced model mice.** (A) An example of gait tracking in wild-type mice. (Ai) An example of gait tracking for 5 minutes out of water. (Aii) An example of gait tracking for 5 minutes in shallow warm water. (Aiii) Bar graphs used to compare gait distance, (Aiv) maximum speed, and (Av) average speed in and out of water in wild-type mice. (B) An example of gait tracking in FM-induced model mice in the second part of the experiment. (Bi) An example of gait tracking for 5 minutes out of water. (Bii) An example of gait tracking for 1 minute in shallow warm water. (Biii) An example of gait tracking for 5 minutes in shallow warm water. (Biv) Bar graphs used to compare gait distance and (Bv) maximum speed during the three gait tests. (Ci) Bar graphs used to compare gait distance and (Cii) maximum speed between wild-type and FM-induced model mice during 5 minutes of free gait in shallow warm water. FM, fibromyalgia.

of water: 18.98 ± 16.37 and 17.66 ± 14.84; Fig 3Biii) (before and after the 5-minute gait in shallow water: 20.19 ±12.94 and 24.00 ± 11.37; Fig 3Biv), and HR irregularity score (before and after the 5-minute gait out of water: 2.81 ± 1.85 and 2.51 ± 1.88; Fig 3Bv) (before and after the 5-minute gait in shallow water: 3.34 ±2.10 and 2.92 ± 1.79; Fig 3Bvi). Further, the HR irregularity score statistically correlated with the SD (Fig 3C). Therefore, only the HR irregularity scores were reported (without the SD) in the results as an evaluation of HR rhythm.

## Cardiac effects of free gait in shallow warm water in FM-induced model mice

The ECG findings were assessed both before and after gait training for the FM-induced mice (Fig 4Ai and 4Aii). However, no significant change was observed in the average HR (beats/min) before and after gait training (before and after the 5-minute gait out of water: 701.14 ± 57.52 and 713.46 ± 45.61; Fig 4Bi) (before and after the 1-minute gait in shallow water: 686.70 ± 52.53 and 681.45 ± 55.58; Fig 4Bii). Although no increase was observed after the 5-minute free gait test out of the water in the HR irregularity score (before and after the 5-minute gait out of water: 2.95 ± 1.59 vs. after 5-minute gait out of water: 3.33 ± 2.41; Fig 4Biii), a significant increase was

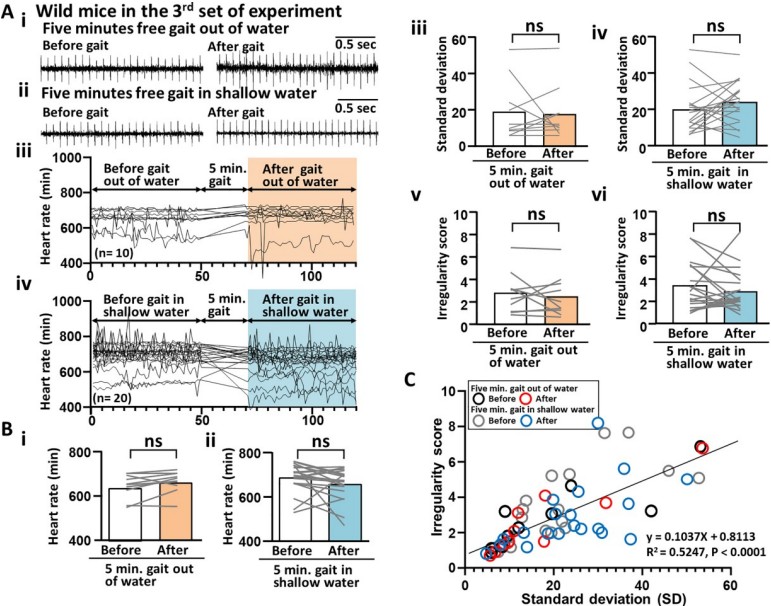

**Fig 3. Cardiac effects of free gait in shallow warm water in wild-type mice.** (A) An example of ECG raw data for wild-type mice. (Left panel of Ai) Before 5 minutes of free gait out of water. (Right panel of Ai) After 5 minutes of free gait out of water. (Left panel of Aii) Before 5 minutes of free gait in shallow warm water. (Right panel of Aii) After 5 minutes of free gait in shallow warm water. (Aiii) A line graph illustrating a change in HR before and after 5 minutes of free gait out of water. (Aiv) A line graph illustrating a change in HR before and after 5 minutes of free gait in shallow warm water. (Bi) Bar graphs comparing the average HR before and after 5 minutes of free gait out of water. (Bii) Before and after 5 minutes of free gait in shallow warm water. (Biii) Bar graphs comparing the SD of the HRs before and after 5 minutes of free gait out of water. (Biv) Before and after 5 minutes of free gait in shallow warm water. (Bv) Bar graphs comparing irregularity score (IS) and HR before and after 5 minutes of free gait out of water. (Bvi) Before and after 5 minutes of free gait in shallow warm water. A graph showing the correlation between SD and IS. ECG, electrocardiogram; HR, heart rate; IS, irregularity score; SD, standard deviation.

observed after the 1-minute free gait test in the shallow warm water (before and after the 1-minute gait in shallow water: $1.88 \pm 0.77$ and $5.57 \pm 4.46$, p<0.05; Fig 4Biv).

## Cardiac effects of long-term free gait in shallow warm water in the FM-induced model mice

The final part of the experiment comprised an investigation of the effects of 5 minutes of free gait in shallow warm water in terms of cardiac function (Fig 5Ai). As shown in Fig 5Aii, after 5 minutes of free gait in shallow warm water, the HR of the FM-induced mice seemingly fluctuated. The HR (beats/min) significantly decreased after 5 minutes of free gait in the shallow warm water (before and after 5-minute gait in shallow water: $688.83 \pm 56.55$ and $622.92 \pm 119.13$, p<0.05; Fig 5Bi). Furthermore, a significant increase was similarly observed in the HR irregularity scores (before and after 5-minute gait in shallow water: $2.50 \pm 1.71$ and $12.17 \pm 8.09$, p<0.05; Fig 5Bii). The average HR (beats/min) after 5 minutes of free gait in shallow warm water was higher than after 1 minute of free gait in shallow warm water (after 5-minute gait in shallow water in wild mice: $671.85 \pm 76.17$, vs. after 5-minute gait out of water in FM model mice: $713.46 \pm 45.61$ vs. after 1-minute gait in shallow water in FM model mice: $681.45 \pm 55.58$ vs. after the 5-minute gait in shallow water in FM model mice: $622.92 \pm 119.13$, p<0.05; Fig 5Ci). The HR irregularity score was also higher after 5 minutes of free gait in shallow warm water than after 1 minute of free gait in shallow warm water (after 5-minute gait in shallow water in wild mice: $3.27 \pm 2.22$ vs. after 5-minute gait out of water in FM model mice:

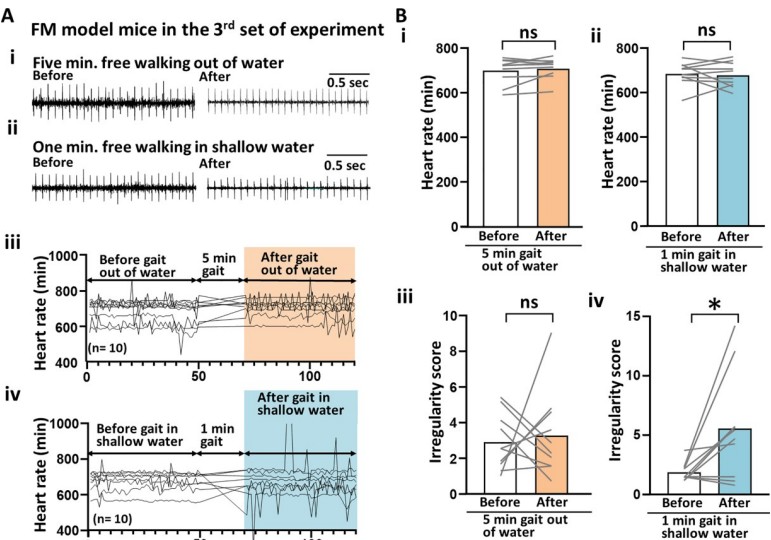

**Fig 4. Cardiac effects of free gait in shallow warm water in the FM-induced model mice.** (A) An example of ECG raw data of FM-induced mice. (Left panel of Ai) Before 5 minutes of free gait out of water. (Right panel of Ai) After 5 minutes of free gait out of water. (Left panel of Aii) Before 1 minute of free gait in shallow warm water. (Right panel of Aii) After 1 minute of free gait in shallow warm water. (Aiii) A line graph illustrating the change in HR before and after 5 minutes of free gait out of water. (Aiv) A line graph illustrating the change in HR before and after 1 minute of free gait in shallow warm water. (Bi) Bar graphs showing a comparison of HRs before and after 5 minutes of free gait out of water. (Bii) Before and after 1 minute of free gait in shallow warm water. (Biii) Bar graphs showing a comparison between IS and HR before and after 5 minutes of free gait out of water. (Biv) Before and after 1 minute of free gait in shallow warm water. ECG, electrocardiogram; FM, fibromyalgia; HR, heart rate; IS, irregularity score.

3.33 ± 2.41 vs. after 1-minute gait in shallow water in FM model mice: 5.57 ± 4.46 vs. after 5-minute gait in shallow water in FM model mice: 12.17 ± 8.09, p<0.05; Fig 5Cii).

## Discussion

This study reported that exercise-induced cardiac arrhythmia was observed at 1-minute gait in shallow water during the acute stage of induced FM in young mice. Both cardiac arrhythmia and a decrease in HR have occurred at 5-minute gait in shallow water at the same mice. However, this phenomenon was not observed in any wild-type mice under any test conditions. Herein, we discuss the proposed mechanisms of prolonged shallow water exercise-induced cardiac arrhythmia in reserpine-injected FM-induced mice, the implication of free gait exercise in shallow warm water, and the clinical applications of the outcomes.

### Mechanisms of prolonged exercise-induced cardiac arrhythmia in reserpine-injected FM-induced model mice

Neuromodulators control cardiac function [37], and neuromodulator imbalances may cause cardiac arrhythmias [38]. For instance, reserpine appears to deplete monoamines, such as noradrenaline, serotonin, and dopamine, in the central and peripheral nervous systems [25]. Therefore, excessive exercise-induced cardiac arrhythmia in the FM-induced mice may be due to arrhythmia derived from the peripheral region [37], the central nervous system [37], or the neuromuscular region [39]. First, the release of neuromodulators, especially noradrenaline and acetylcholine, regulates cardiac function [37, 38]. Therefore, such exercise may have over-stressed the FM-induced mice and caused the cardiac arrhythmia due to a depletion in the level of monoamines needed for cardiac function. Second, it has been reported that monoamines released in the rostral

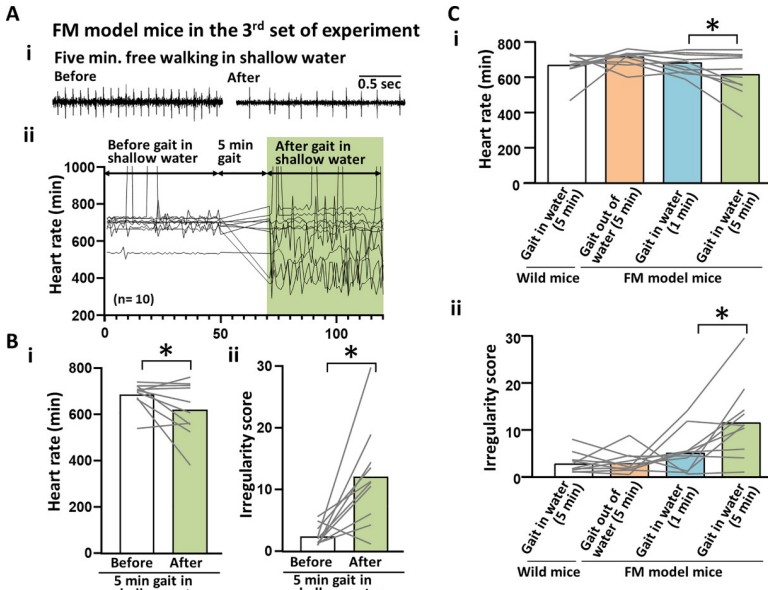

**Fig 5. Cardiac effects of long-term free gait in shallow warm water in the FM-induced model mice.** (A) An example of ECG raw data involving FM-induced model mice. (Left panel of Ai) Before 5 minutes of free gait in shallow warm water. (Right panel of Ai) After 5 minutes of free gait in shallow warm water. (Aii) A line graph illustrating the change in HR before and after 5 minutes of free gait in shallow warm water. Bar and line graphs comparing the average HR before and after 5 minutes of free gait in shallow warm water. (Bii) Bar and line graphs are comparing the IS of the HR before and after 5 minutes of free gait in shallow warm water. (C) Bar and line graphs are comparing both HR (Ci) and the IS of the HR (Cii) under the four experimental conditions. After 5 minutes of free gait in shallow water in wild-type mice (gait in water [5 min]). After 5 minutes of free gait out of water in FM-induced model mice (gait out of water-gait [5 min]). After 1 minute of free gait in shallow warm water in FM-induced model mice (gait in water [1 min]). After 5 minutes of free gait in shallow warm water in FM-induced model mice (gait in water [5 min]). ECG, electrocardiogram; FM, fibromyalgia; HR, heart rate; IS, irregularity score.

ventrolateral medulla (of the medulla oblongata) play a crucial role in the control of autonomic function, such as in respiratory and cardiac regulation [37, 40–42]. Therefore, monoamine depletion in the medulla might equally result in cardiac arrhythmia. Third, monoaminergic neurotransmission in the cortical motor area also might be downregulated [37]. Fourth, the neuromuscular region may be indirectly involved. The FM model mice showed a temporary decrease in weight (Fig 1A), which may have been due to muscle atrophy because of pain and reduced movement [39]. Acetylcholine is only released at the neuromuscular junction, as it is not a monoamine [43]. However, skeletal muscles are controlled by motor neurons in the ventral horn of the spinal cord. It is reported that noradrenaline, one of the monoamines, excites motor neurons [44], suggesting that changes in monoamine release in the ventral horn of the spinal cord may modulate the excitability of the motor neurons, which may downregulate the acetylcholine release in neuromuscular junction [44]. Further, FM patients are often afflicted with fatigue syndrome [4]. Muscle atrophy and fatigue may also be the cause for the decrease in body weight of FM-induced mice after reserpine injection that was observed in this study (Fig 1A). Therefore, excessive gait might overload the heart muscles during the acute stage in FM mice.

## Implications of free gait in shallow warm water as a form of exercise for FM-induced mice

In the investigation of the free gait of FM-induced mice in shallow warm water, consideration was given to the following: a short-duration free gait, cardiovascular issues, the use of

inexpensive equipment, and the simplest and easiest method of assessment. While a free gait in shallow warm water enabled an easy assessment of distance covered and average and maximum speeds, the risk of cardiac arrhythmia remained, especially during the long-duration 5-minute exercise of the FM-induced model mice in the acute stage of this disease. While a treadmill that assesses passive gait in the FM-induced model mice may be inaccessible and expensive, the cheaper treadmill alternative from the pet shop cannot precisely control the speed. On the contrary, short-term free gait sessions in shallow warm water were found to be an inexpensive and easy way for assessing the FM-induced mice. Further, the method used in this study allowed us to perform both the exercise and its evaluation of multiple mice in one cage at the same time.

## Study limitations and clinical applications

This study had several limitations. First, HR was measured for only 20 seconds. If the HR had been continuously monitored, the relationship between exercise load, gait time, and cardiac arrhythmia could have been more clearly determined in the FM-induced mice. Second, the study had a critical experimental limitation. Although cardiac abnormalities were detected in ECG recordings acquired after the gait test, owing to the difficulty in performing ECG recordings during the gait tests, we could not determine the starting point of the abnormality nor if it began during the gait test. Third, we could not precisely analyze the ECG recordings, such as QRST wave analysis, acquired from the mice since the QRST wave is unclear in the ECG signals from the mice (S1D Fig). Therefore, we also could not specify which regions of the heart were normal or abnormal.

While there is a study showing that FM patients have similar cardiovascular responses to submaximal exercise as healthy control subjects [45], other new studies reported that FM patients had higher HR [27] and impaired cardiac function [28]. Further, they seem to have a higher risk of coronary heart disease-related events [29] and delayed HR recovery after a treadmill graded gait exercise [46]. However, it could not be clear how these cardiovascular problems in FM patients relate to the exercise-induced cardiac arrhythmia and decrease of HR in FM model mice. Further, we do not know how cardiovascular problems aggravate disease progression. Inevitably, previous studies reported that moderate aerobic exercise reduces both musculoskeletal pain and autonomic dysfunction for FM patients [17, 47–49]. Therefore, medical staff should be aware of the following three matters. First, FM patients have both musculoskeletal pain and cardiovascular risks. Second, the load and the time spent performing therapeutic exercises for FM patients should be precisely arranged, with initially low and short duration; otherwise, unexpected cardiovascular events may occur during exercise in these patients. Third, when FM patients perform therapeutic exercises, ECG or HR monitoring should be considered.

## Conclusion

This study reported a prolonged and excessive exercise-induced cardiac abnormality involving a decrease in the HR and the occurrence of cardiac arrhythmia in the acute stage of FM-induced mice. Although a short-term free gait in shallow warm water may be advantageous for increasing the motor activity of FM-model mice, our data indicates it can be associated with changes in heart rate and even arrhythmias. Physiotherapists and other health professionals should be aware of these potential risks when considering strenuous exercise as a treatment in FM patients. We suggest a gradual increase in exercise duration may be warranted.

## Supporting information

**S1 Fig. Experimental protocols and flow charts of the experiments.** (A) The experimental protocol to produce FM-induced mice. (B) A flow chart of the first experiment to measure body weight and rectal temperature, and video recordings of free gait inside the cage. (C) A flow chart of the second experiment with video recordings before and after the three types of gait tests. (D) Raw data from ECG recordings taken to evaluate cardiac function. R means a negative peak of wave in the ECG. (E) A flow chart of the third experiment for ECG recordings before and after the three types of gait tests. ECG, electrocardiogram; FM, fibromyalgia. (TIF)

## Acknowledgments

An illustration of the black mouse in Fig 1B, has been cited from © 2016 DBCLS TogoTV (https://togotv.dbcls.jp/).

## Author Contributions

**Conceptualization:** Atsushi Doi, Megumu Yoshimura, Min-Chul Shin.

**Data curation:** Taiki Nakata.

**Formal analysis:** Taiki Nakata, Atsushi Doi, Daisuke Uta.

**Funding acquisition:** Atsushi Doi.

**Investigation:** Atsushi Doi.

**Methodology:** Taiki Nakata, Daisuke Uta.

**Project administration:** Atsushi Doi, Megumu Yoshimura, Min-Chul Shin.

**Resources:** Atsushi Doi.

**Software:** Taiki Nakata, Daisuke Uta.

**Supervision:** Daisuke Uta, Megumu Yoshimura, Min-Chul Shin.

**Validation:** Megumu Yoshimura, Min-Chul Shin.

**Visualization:** Megumu Yoshimura, Min-Chul Shin.

**Writing – original draft:** Taiki Nakata, Daisuke Uta, Megumu Yoshimura, Min-Chul Shin.

**Writing – review & editing:** Taiki Nakata, Atsushi Doi, Daisuke Uta, Megumu Yoshimura.

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
