## [Decision Letter · Decision Letter 0]

25 Jun 2020

PONE-D-20-16502

A long and excessive exercise-induced abnormal cardiac arrhythmia in young model mice with acute stage fibromyalgia

PLOS ONE

Dear Dr. Doi,

Thank you for submitting your manuscript to PLOS ONE. After careful consideration, we feel that it has merit but does not fully meet PLOS ONE’s publication criteria as it currently stands. Therefore, we invite you to submit a revised version of the manuscript that addresses the points raised during the review process.

We look forward to receiving your revised manuscript.

Kind regards,

Etsuro Ito

Academic Editor

PLOS ONE

Journal Requirements:

Additional Editor Comments (if provided):

The two independent reviewers basically like your manuscript, and so please revise it according to their comments,

Reviewers' comments:

Reviewer's Responses to Questions

**Comments to the Author**

1. Is the manuscript technically sound, and do the data support the conclusions?

Reviewer #1: Yes

Reviewer #2: Yes

2. Has the statistical analysis been performed appropriately and rigorously? 

Reviewer #1: Yes

Reviewer #2: Yes

3. Have the authors made all data underlying the findings in their manuscript fully available?

Reviewer #1: Yes

Reviewer #2: No

4. Is the manuscript presented in an intelligible fashion and written in standard English?

Reviewer #1: Yes

Reviewer #2: No

5. Review Comments to the Author

Reviewer #1: The papers described that a prolonged and excessive exercise induced cardiac abnormality involving a decrease in heart rate and the occurrence of cardiac arrhythmia during the acute stage of induced fibromyalgia (FM) in model mice. This phenomenon was not observed in any of the wild-type mice under any of the test conditions.

This research is potentially interesting, however, following points should be addressed.

1. Has this paper been proofread by native speakers?

2. In the abstract, “Results: the cardiac” should be corrected as “Results: The cardiac”.

3. In the text, references 16-19 are missing from the text.

4. “In general, medications (Ferreira-Dos-Santos et al., 2018; Sarmento et al., 2019),”

A quote of above references is mistaken. Please use numeric references.

5. For the instruments or reagent, model number, company name, city, province (if USA), country should be included.

For example,

camera (100 frames/sec, TZ-35, Panasonic, Japan)

digitizer (Axon DigiData 1322, USA)

6. How the mice were fixed a position for measurements of the ECG signals?

7. Authors utilized following formula, Sn = 100*ABS (Pn – Pn-1)/Pn-1, for analysis of fluctuation of the ECG signals.

Is it a standard way to observe the variation of the ECG signals?

8. Mice generally do not like walking or swimming in the water; even depth is pretty shallow.

Why did you come up with that idea?

9. What is the purpose of measuring rectal temperature in FM animals, as there is nothing in the discussion of rectal temperature?

Was there any previous study on temperature changes?

10. Have you already tried walking in a shallow pool as a stimulus or exercise for FM mice?

If so, what is the effect?

11. The authors described four mechanisms caused by monoamine depletion as a mechanism of excessive exercise-induced cardiac arrhythmias in FM-induced model mice injected with reserpine.

What do you think about other mechanisms besides monoamine depletion for excessive exercise-induced cardiac arrhythmia, as the potential mechanism of fibromyalgia symptoms is not yet known?

12. In the Fig. 1A, “(Days after 3td reserpine injection)” should be corrected as “(Days after 3rd reserpine injection)”.

13. In the Fig. 2A, why did the bodyweight of FM mice decrease right after first shot of the reserpine?

Then, are there any changes in the behavior of FM model animals in day and night time?

14. In the reference list, following journals should be corrected.

43. Kanda Y. Investigation of the freely available easy-to-use software {’EZR’} for medical statistics. Bone Marrow Transplant. 2013;48: 452–458. should be corrected as

43. Kanda Y. Investigation of the freely available easy-to-use software ’EZR’ for medical statistics. Bone Marrow Transplant. 2013;48: 452–458.

49. Beech DJ. Actions of neurotransmitters and other messengers on Ca2+ channels and K+ channels in smooth muscle cells. Pharmacol Ther. 1997/01/01. 1997;73: 91–119. doi:10.1016/s0163-7258(97)87271-3, should be corrected as

Beech DJ. Actions of neurotransmitters and other messengers on Ca2+ channels and K+ channels in smooth muscle cells. Pharmacol Ther. 1997/01/01. 1997;73: 91–119. doi:10.1016/s0163-7258(97)87271-3.

Reviewer #2: The authors use chronic reserpine treatment for 1 week to induce cardiac arrhythmias, that manifest as irregular HR upon brief 1 minute walking in warm water, and that become exacerbated (with reduced mean HR) when walking in warm water is prolonged (5 mins). As reserpine dosing has been previously shown to induce fibromyalgia (FM) symptoms in mice, and as human FM patients have cardiac abnormalities, this new finding adds an additional aspect to this FM model that may have utility in researching mechanisms or treatments for FM. In particular, the approach is readily accessible with minimal costs, which can broaden the utility of this model.

The data is clear in the Figures, and appropriately presented and analyzed. The experiments appear robust and well designed. The Figures are very clear.

However the integration of the results with the current literature in the conclusions, and the rationale and introduction , and the way the methods is described needs significant work. The results do not seem to be testing or reporting any FM treatment approach (as stated, for example, in abstract) and the fact that the reserpine treated mouse has been previously validated as a FM model is not mentioned (in fact, it is stated reserpine is a treatment?). What has been shown is that this model induces cardiac abnormalities, and this should be the focus of Intro and Discussion.

A key point therefore is to look further into the ECG pattern and describe (and show) what the abnormalities are that cause the irregularity. I think the data should already be there.

There are many areas in the text that need clarification. The English is good, but the authrors just need to be more concise with details, rationale and conclusion. I have many suggestions below (sorry) but I hope it helps a bit and improves the paper.

Abstract

Background is wrong. Study doesn’t aim to test a treatment. Its investigating CVS effects of the reserpine-FM model. Methods: Last sentence unclear, seems a conclusion, not a method, but then seems to contradict result? Just leave out

Ethics – anaesthesia not mentioned.

Data – Some misunderstanding I think. Data stated as not available but no real reasons given? Says all data in ms, but individual values not in ms?

Title “Abnormal” cardiac arrhythmia. By definition, arrhythmias are abnormal. Is this needed? Delete? And does long and excessive refers to exercise, not arrhythmia? I suggest something like “Identification of exercise induced cardiac arrhythmia in fibromyalgia model mice”

Intro.

- Ref 14 describes that the reserpine model has been described quite thoroughly as a model of FM. So this needs to be sated and the features described in the Introduction. SO the rationale is then to evaluate exercise to reduce cardiac abnormalities associated with this model? Then one needs to 1st characterize these cardiac abnormalities (or other features of FM such as allodynia), then test exercise against these control features. And the study rationale made clearer. Its unclear whether cardiac abnormalities found before.

L92. Suggests reserpine used to reduce FM symptoms (line 92), but you are using to produce FM symptoms. Rewrite to clarify.

L10 9. Is anything known of cardiac symptoms in FM mice? Has this been seen before? This is an important point, as the paper may be the first to identify this so should be clarified in the introduction. Why is it stated a “possibility of cardiovascular complications”.

And treadmills are available for mice – cheaply in pet shops - delete this.

Methods.

L130. Reserpine 1ml/kg. What dose in mg/kg? And 1ml/kg is about 0.02 ml in a 20g mice. Check this. Can you achieve such low volumes accurately? What type of needle?

How did you do “out of the water” and “shallow water”. Were the mice taken from home cages to the plastic cage with warm water? Was the out of water the same procedure (ie taken from home cage to plastic cage) but without the water. Was every mice given the three tests? Was it wiped to clear any residual scent between tests? How did the gait software work – via some digital tracking?

Were these animals anesthetized to place the electrodes and for rectal temperature? Please give details

ECG. Seems nice. How was the period measured? (eg between peak of QRS complex). many cycles used to obtain a mean irregularity score?

In general, Figure 1 describes the experiments and cohorts well. Use this when your going through the animal numbers.

Results.

Figures are great – very clearly show methods protocol. Results clearly described

Was the irregularities related to the distance travelled? For instance, they were seen with 1 min warm water but not 5 mins out of water. Was the distance and speed (ie exertion) greater in 1 min warm water?

The mice lost a lot of weight. Figures say 7 days after reserpine, but weight graphs only go out to 5 or 6 days after reserpine?

What causes the irregularity? Is it missed or delayed beats? Can the authors look at the ECG and analyze what the specific irregularity(ies) is/are?. Does this relate to human FM abnormalities?

Discussion

What the paper shows is that it extends the reserpine FM model to show that cardiovascular abnormalities occur with 1 and 5 min warm water exercise, being an increase in irregular rhythm and, for 5 mins, a decrease in HR. This should be stated clearly in the context of whether FM models have seen cardiovascular parameters, how it relates to human FM and what may be the nature and cause of the irregularity.

Is this interpreted as an exercise effect or a stress-effect, or some combination?

Some discussion of what irregularity means. Is this abnormalities in specific regions or the QRST waves (Q-T delay for example) or is ECG shape normal?

P280-281 RVM “monomaines play a crucial role” – in what? How precisely? This sounds interesting and maybe relevant

285-290. It seems an ischemic effect postulated due to poor blood flow regulation. Cardiac blood flow largely autoregulation isn’t it, rather than Noradrenalin? I don’t see how neuromuscular effects or cardiac blood flow could be involved. Most arhhythmas due to deficits in cellular control of excitability, so 1st need to identify the nature of the “irregularity” then start looking towards possible effects on cardiac channels or they posttranslational regulation

4.2 makes an excellent point of this is a simple design with cheap, accessible methods. I think a mouse exercise wheel is also cheap from a pet shop, and one can set for one direction and count revolutions over time.

Conclusion. Even 1 min warm water gait had increased irregularity, so cant conclude this is safe and “beneficial”, although I appreciate it did increase movement (due to fear or stress?). So conclusion should rewrite.

6. PLOS authors have the option to publish the peer review history of their article (what does this mean?). If published, this will include your full peer review and any attached files.

Reviewer #1: No

Reviewer #2: No

---

## [Author Response · Author response to Decision Letter 0]

18 Aug 2020

PLOS ONE's suggestion

1. Please ensure that your manuscript meets PLOS ONE's style requirements, including those for file naming. The PLOS ONE style templates can be found at https://journals.plos.org/plosone/s/file?id=wjVg/PLOSOne_formatting_sample_main_body.pdf

Answer for PLOS ONE's suggestion

We have revised the manuscript and figures in accordance with the PLOS ONE style templates.

Additional Editor Comments:

The two independent reviewers basically like your manuscript, and so please revise it according to their comments,

Answer for additional Editor Comments:

We have uploaded our figure files to the PACE digital diagnostic tool.

 

Reviewer's Responses to Questions

Question #1:

Is the manuscript technically sound, and do the data support the conclusions?

Reviewer #1: Yes

Reviewer #2: Yes

Response:

We thank the reviewers for this feedback.

Question #2:

Has the statistical analysis been performed appropriately and rigorously?

Reviewer #1: Yes

Reviewer #2: Yes

Response:

We thank the reviewers for this feedback.

Question #3:

Have the authors made all data underlying the findings in their manuscript fully available?

Reviewer #1: Yes

Reviewer #2: No

Response:

We have included all data underlying the findings in our manuscript to ensure the data is fully available. 

Question #4:

Is the manuscript presented in an intelligible fashion and written in standard English?

Reviewer #1: Yes

Reviewer #2: No

Response:

The manuscript has been reviewed and revised by Editage.

Question #5:

PLOS authors have the option to publish the peer review history of their article (what does this mean?). If published, this will include your full peer review and any attached files.

Do you want your identity to be public for this peer review? For information about this choice, including consent withdrawal, please see our Privacy Policy.

Reviewer #1: No

Reviewer #2: No

Response:

We thank the reviewers for their peer review.

Reviewer’s Comments to the Author

Reviewer #1: 

The papers described that a prolonged and excessive exercise induced cardiac abnormality involving a decrease in heart rate and the occurrence of cardiac arrhythmia during the acute stage of induced fibromyalgia (FM) in model mice. This phenomenon was not observed in any of the wild-type mice under any of the test conditions.　This research is potentially interesting, however, following points should be addressed.

Comment 1: Has this paper been proofread by native speakers?

Response: The manuscript has been reviewed and revised by Editage.

Comment 2: In the abstract, “Results: the cardiac” should be corrected as “Results: The cardiac”.

Response: We have revised the results in the abstract (line 46).

Comment 3: In the text, references 16-19 are missing from the text.

Response: We have ensured that references 16–19 are appropriately cited within the text (line 61) and at the end of the manuscript (Lines 493-499). Furthermore, a couple of references have already been omitted.

Comment 4: “In general, medications (Ferreira-Dos-Santos et al., 2018; Sarmento et al., 2019),”

A quote of above references is mistaken. Please use numeric references.

Response: We have ensured that all the references are listed at the end of manuscript and are numbered in the order in which they appear in the text. We have also ensured that all in-text citations are listed as numbers within square brackets in accordance with PLOS ONE’s formatting requirements.

Comment 5: For the instruments or reagent, model number, company name, city, province (if USA), country should be included.

For example,

camera (100 frames/sec, TZ-35, Panasonic, Japan)

digitizer (Axon DigiData 1322, USA)

Response: We have included the model number, company name, city, and country for all commercial sources of instruments and reagents (lines 105,147,148, 152, 164, 176).

Comment 6: How the mice were fixed a position for measurements of the ECG signals?

Response: We have added a description to clarify that the mice were gently held in the prone position by the examiner (Line 176).

Comment 7: Authors utilized following formula, Sn = 100*ABS (Pn – Pn-1)/Pn-1, for analysis of fluctuation of the ECG signals.

Is it a standard way to observe the variation of the ECG signals?

Response: The irregularity score (IS) is often used to evaluate respiratory rhythm. However, the IS might not be standard in the analysis of ECG signals. In this study, since the heartbeat of the mice was fast, we decided to evaluate the fluctuation of the heartbeat. Therefore, in this study, we utilized the IS of the respiratory rhythm to evaluate the regularity of the heartbeat. Further, a recent study used a similar analysis method to evaluate the R-R interval (Karey E, Pan S, Morris AN, Bruun DA, Lein, PJ, Chen, C. The use of percent change in RR interval for data exclusion in analyzing 24-h time domain heart rate variability in rodents. Front. Physiol. 2019;10:693)

Comment 8: Mice generally do not like walking or swimming in the water; even depth is pretty shallow. Why did you come up with that idea?

Response 8: We have included a description in the Introduction section to clarify that we used underwater walking as a method to decrease overload on the antigravity muscles (line 64).

Comment 9: What is the purpose of measuring rectal temperature in FM animals, as there is nothing in the discussion of rectal temperature?

Was there any previous study on temperature changes?

Response: We initially thought that reserpine related monoamines downregulation may change the rectal temperature since the body temperature is controlled by the autonomic function as a phenomenon of the monoamines downregulation. However, the changing of the temperature was temporary, and we could not scientifically clear the reason about the temporal change of the temperature. Further, we have included a description in the Material and Methods section that clarifies that measuring rectal temperature was used as method to measure body temperature in the mice. We have also included a reference to a prior study that used this method (reference 31). The change in the rectal temperature was negligible and thus, we did not expand further on the rectal temperature of the mice in the Discussion section. 

Comment 10: Have you already tried walking in a shallow pool as a stimulus or exercise for FM mice? If so, what is the effect?

Response: We have performed these experiments; however, as it is currently under peer-review we cannot describe the results in great detail. Underwater walking does not appear to have an effect on the pain threshold. 

Comment 11: The authors described four mechanisms caused by monoamine depletion as a mechanism of excessive exercise-induced cardiac arrhythmias in FM-induced model mice injected with reserpine.

What do you think about other mechanisms besides monoamine depletion for excessive exercise-induced cardiac arrhythmia, as the potential mechanism of fibromyalgia symptoms is not yet known?

Response: The Discussion has been revised to include further detail about how fatigue and muscle atrophy may be related to a decrease in weight in fibromyalgia patients (line 412). It is possible that excessive gait may overload the heart muscles, and not skeletal muscles, in acute stage FM mice.

Comment 12: In the Fig. 1A, “(Days after 3td reserpine injection)” should be corrected as “(Days after 3rd reserpine injection)”.

Response: We have revised this title accordingly in Supplemental Figure 1A (S1 FigureA). 

Comment 13: In the Fig. 2A, why did the bodyweight of FM mice decrease right after first shot of the reserpine?

Then, are there any changes in the behavior of FM model animals in day and night time?

Response: Although Nagakura et al. reported a decrease in weight [31], they do not describe a mechanism underlying the weight decrease. Skeletal muscle atrophy should not appear in reserpine injected FM-induced model mice. However, it is still currently unclear why the reserpine injection decreases their weight. 

After the reserpine injection, our data show that the locomotive activity of FM model animals is inhibited both during the day time and night time.

Comment 14: In the reference list, following journals should be corrected.

43. Kanda Y. Investigation of the freely available easy-to-use software {’EZR’} for medical statistics. Bone Marrow Transplant. 2013;48: 452–458. 

49. Beech DJ. Actions of neurotransmitters and other messengers on Ca2+ channels and K+ channels in smooth muscle cells. Pharmacol Ther. 1997/01/01. 1997;73: 91–119. doi:10.1016/s0163-7258(97)87271-3, should be corrected as

Response: We have revised a new reference 36, and omitted original reference 49. 

 (line 615). 

 

Reviewer #2: 

The authors use chronic reserpine treatment for 1 week to induce cardiac arrhythmias, that manifest as irregular HR upon brief 1 minute walking in warm water, and that become exacerbated (with reduced mean HR) when walking in warm water is prolonged (5 minutes). As reserpine dosing has been previously shown to induce fibromyalgia (FM) symptoms in mice, and as human FM patients have cardiac abnormalities, this new finding adds an additional aspect to this FM model that may have utility in researching mechanisms or treatments for FM. In particular, the approach is readily accessible with minimal costs, which can broaden the utility of this model.

The data is clear in the Figures, and appropriately presented and analyzed. The experiments appear robust and well designed. The Figures are very clear.

However the integration of the results with the current literature in the conclusions, and the rationale and introduction, and the way the methods is described needs significant work. The results do not seem to be testing or reporting any FM treatment approach (as stated, for example, in abstract) and the fact that the reserpine treated mouse has been previously validated as a FM model is not mentioned (in fact, it is stated reserpine is a treatment?). What has been shown is that this model induces cardiac abnormalities, and this should be the focus of Intro and Discussion. A key point therefore is to look further into the ECG pattern and describe (and show) what the abnormalities are that cause the irregularity. I think the data should already be there. There are many areas in the text that need clarification. The English is good, but the authors just need to be more concise with details, rationale and conclusion. I have many suggestions below (sorry) but I hope it helps a bit and improves the paper.

Comment 1: Abstract

Background is wrong. Study doesn’t aim to test a treatment. It’s investigating CVS effects of the reserpine-FM model. 

Response: The Abstract has been revised to clarify that our aim was to investigate the cardiac effect of a prolonged shallow water gait in an FM-induced mouse model (line 37).

Comment 2: Methods: Last sentence unclear, seems a conclusion, not a method, but then seems to contradict result? Just leave out

Response: We have removed the last sentence of the Methods described in the Abstract in order to improve clarity (line 45). 

Comment 3: Ethics – anesthesia not mentioned.

Response: We have not mentioned anesthesia in the Ethics section since we did not use anesthesia for mice when injecting reserpine or during ECGs.

Comment 4: Data – Some misunderstanding I think. Data stated as not available but no real reasons given? Says all data in manuscript, but individual values not in manuscript?

Response: We have revised this and included all experimental data in the manuscript (lines 214, 232, 238, 271, 293, 307, 311, 340, 352).

Comment 5: Title “Abnormal” cardiac arrhythmia. By definition, arrhythmias are abnormal. Is this needed? Delete? And does long and excessive refers to exercise, not arrhythmia? I suggest something like “Identification of exercise induced cardiac arrhythmia in fibromyalgia model mice”

Response: We thank the reviewer for this suggestion of the title. We have deleted “a long” and “abnormal” from the title to remove the redundancy (line 1). The new title is “Excessive exercise induces cardiac arrhythmia in a young fibromyalgia mouse model”. 

Comment 6: Introduction

- Ref 14 describes that the reserpine model has been described quite thoroughly as a model of FM. So this needs to be stated and the features are described in the Introduction. Thus, the rationale is then to evaluate exercise to reduce cardiac abnormalities associated with this model? Then one needs to 1st characterize these cardiac abnormalities (or other features of FM such as allodynia), then test exercise against these control features. And the study rationale made clearer. It’s unclear whether cardiac abnormalities found before.

Response: We have revised the Introduction to describe the features of reserpine model. We have also described the cardiac abnormalities associated with this model and clarified the rationale of using exercise to reduce these cardiac abnormalities (starting on line 58).

Comment 7: Original line 92 suggests reserpine used to reduce FM symptoms (line 92), but you are using to produce FM symptoms. Rewrite to clarify.

Response: We have edited this to clarify that we used reserpine to produce FM symptoms (line 74).

Comment 8: Original line109 

Is anything known of cardiac symptoms in FM mice? Has this been seen before? This is an important point, as the paper may be the first to identify this so should be clarified in the introduction. Why is it stated a “possibility of cardiovascular complications”.

Response: In a search conducted on the search engine PubMed using the keywords “fibromyalgia” and “cardiac arrhythmia” (on July 10, 2020), we found 26 reports. However, when we used the keywords “fibromyalgia”, “cardiac arrhythmia”, and “animal”, “rat”, or “mouse”, there were no studies found. Therefore, we have edited the Introduction to highlight that there are no known studies that evaluate exercise-induced cardiac arrhythmia in an FM-induced mouse model (line 86). We have also clarified our description of the cardiovascular complications in the Introduction by removing the word “possibility” (line 84).

Comment 9: And treadmills are available for mice – cheaply in pet shops – delete this.

Response: We have removed the sentence about treadmills in the Introduction.

Comment 10: Methods.

Original line 130. 

Reserpine 1ml/kg. What dose in mg/kg? And 1ml/kg is about 0.02 ml in a 20g mice. Check this. Can you achieve such low volumes accurately? What type of needle?

Response: We have revised the description of the dosage to 10 mL/kg and clarified that this would be 0.3 mL of reserpine for a mouse with a body weight of 30 g (line 111).

Comment 11: How did you do “out of the water” and “shallow water”. Were the mice taken from home cages to the plastic cage with warm water? Was the out of water the same procedure (ie taken from home cage to plastic cage) but without the water. Was every mice given the three tests? Was it wiped to clear any residual scent between tests? 

Response: We used different cages for breeding and the experiments. This included cages with shallow water and cages without water. We have revised our description of the gait test to clarify the method used with the cages (starting at line 128). 

Comment 12: How did the gait software work – via some digital tracking?

Response: We have revised our description of the gait software (line 167).

Comment 13: Were these animals anesthetized to place the electrodes and for rectal temperature? Please give details

Response: No, the mice were not anesthetized before we conducted the ECG and measured the rectal temperature. Thus, we have excluded this from our description of measurement of the rectal temperature (line 147). 

Comment 14: The measurements of ECG recordings seem nice. How was the period measured? (eg between peak of QRS complex). How many cycles used to obtain a mean irregularity score?

Response: We have revised our description of our measurements of ECG recordings in the Material and Methods section (line 170). We detected ECG signals before and after the gait tests, and the ECG signals were recorded for at least 20 seconds (200–250 cycles were recorded on average for each mouse). The HR irregularity score was determined using 50 ECG cycles.

Comment 15: In general, Figure 1 describes the experiments and cohorts well. Use this when you are going through the animal numbers.

Response: We have changed Fig. 1 to Supplemental Fig. 1.

Comment 16: Results.

Figures are great – very clearly show methods protocol. Results clearly described

Response: We thank the reviewer for this feedback.

Comment 17: Was the irregularities related to the distance travelled? For instance, they were seen with 1 min warm water but not 5 min out of water. Was the distance and speed (ie exertion) greater in 1 min warm water?

Response: We believe that the HR irregularity may relate to the amount of activity, such as gait distance. We think that excessive activity might overload the cardiac muscles and induce cardiac arrhythmia. 

We have revised Fig. 2B to show the increase in both gait distance and speed in an individual mouse.

Comment 18: The mice lost a lot of weight. Figures say 7 days after reserpine, but weight graphs only go out to 5 or 6 days after reserpine?

Response: We have revised “seven days” to “five and six days” in new S1 Figure A. 

Comment 19: What causes the irregularity? Is it missed or delayed beats? Can the authors look at the ECG and analyze what the specific irregularity (ies) is/are? Does this relate to human FM abnormalities?

Response: We think that the ECG irregularity observed in the mice has two patterns. The first pattern is a delayed or skipped ECG, meaning the distance between two ECG cycles is long. The second pattern is a doublet-like ECG, meaning the distance between the two ECG cycles is short. These two components are often mixed. We think that these ECG abnormalities patterns are related to human ECG abnormalities.

Comment 20: Discussion

What the paper shows is that it extends the reserpine FM model to show that cardiovascular abnormalities occur with 1 and 5 min warm water exercise, being an increase in irregular rhythm and, for 5 minutes, a decrease in HR. This should be stated clearly in the context of whether FM models have seen cardiovascular parameters, how it relates to human FM and what may be the nature and cause of the irregularity. Is this interpreted as an exercise effect or a stress-effect, or some combination?

Response: 

We clearly put results in content of both results and the beginning of discussion, which reviewer asked to us (starting at line 305, 336 and 377). However, we also explained that it could not be clear how these cardiovascular problems in FM patients relate to the exercise-induced cardiac arrhythmia and decrease of HR in FM model mice. Further, we wrote unknown about how cardiovascular problems aggravate disease progression. (starting at line 452). We are thinking both irregular rhythm and a decrease in HR caused by 5 minutes of warm water exercise mainly originated as a result of the exercise effect. However, we also cannot perfectly exclude the possibility of the exercise-induced stress effect.

Comment 21: Some discussion of what irregularity means. Is this abnormalities in specific regions or the QRST waves (Q-T delay for example) or is ECG shape normal?

Response: Our ECG analysis was based on both the numbers and regularity of detected “R” in the ECG (line 187, see S1 Fig D). We could not precisely analyze the QRST waves, such as the Q-T delay, since the QRST wave was unclear in the ECG signals from the mice. Therefore, we cannot say which parts of the heart are normal or abnormal. We have revised our description of the ECG signals to include this information (line 443).

Comment 22: Original line 280-281 

RVM “monomaines play a crucial role” – in what? How precisely? This sounds interesting and maybe relevant

Response: We thank the reviewer for this pertinent comment. Monoamines have been reported to play a crucial role in the control of autonomic function, including respiratory and cardiac regulation. We have added this additional information with supporting references (line 399; references 37, 40–42).

Comment 23: Original line 285-290. 

It seems an ischemic effect postulated due to poor blood flow regulation. Cardiac blood flow largely autoregulation isn’t it, rather than Noradrenalin? I don’t see how neuromuscular effects or cardiac blood flow could be involved. Most arrhythmia due to deficits in cellular control of excitability, so 1st need to identify the nature of the “irregularity” then start looking towards possible effects on cardiac channels or they posttranslational regulation

Response: We agree that cardiac blood flow is largely a result of autoregulation. Therefore, we have revised our Discussion section to address this (line 407).

Comment 24: 4.2 makes an excellent point of this is a simple design with cheap, accessible methods. I think a mouse exercise wheel is also cheap from a pet shop, and one can set for one direction and count revolutions over time.

Response: We have revised our Discussion section to discuss how treadmills are also an inexpensive and accessible method, but precise control of the speed may be a limitation (line 426).

Comment 25: Conclusion. Even 1 min warm water gait had increased irregularity, so cannot conclude this is safe and “beneficial”, although I appreciate it did increase movement (due to fear or stress?). So conclusion should rewrite.

Response: We have revised our Conclusion to address this point (starting on line 466). We have clarified that although a short-term free gait test in shallow warm water may be advantageous for increasing the activity of FM-induced mice, we should be aware of the risk of prolonged and excessive exercise-induced cardiac arrhythmia. For gait exercises in shallow water with FM-induced mice, there should be a gradual increase in duration.

---

## [Decision Letter · Decision Letter 1]

8 Sep 2020

Excessive exercise induces cardiac arrhythmia in a young fibromyalgia mouse model

PONE-D-20-16502R1

Dear Dr. Doi,

We’re pleased to inform you that your manuscript has been judged scientifically suitable for publication and will be formally accepted for publication once it meets all outstanding technical requirements.

Kind regards,

Etsuro Ito

Academic Editor

PLOS ONE

Additional Editor Comments (optional):

Thank you for submitting your wonderful manuscript to PLOS ONE.

Reviewers' comments:

Reviewer's Responses to Questions

**Comments to the Author**

1. If the authors have adequately addressed your comments raised in a previous round of review and you feel that this manuscript is now acceptable for publication, you may indicate that here to bypass the “Comments to the Author” section, enter your conflict of interest statement in the “Confidential to Editor” section, and submit your "Accept" recommendation.

Reviewer #1: All comments have been addressed

Reviewer #2: All comments have been addressed

2. Is the manuscript technically sound, and do the data support the conclusions?

Reviewer #1: Yes

Reviewer #2: Yes

3. Has the statistical analysis been performed appropriately and rigorously? 

Reviewer #1: Yes

Reviewer #2: Yes

4. Have the authors made all data underlying the findings in their manuscript fully available?

Reviewer #1: Yes

Reviewer #2: Yes

5. Is the manuscript presented in an intelligible fashion and written in standard English?

Reviewer #1: Yes

Reviewer #2: Yes

6. Review Comments to the Author

Reviewer #1: The authors answered the question appropriately.

This manuscript is well revised, and it is acceptable in the present form.

Reviewer #2: All comments adequately addressed. Well Done, paper reads much better

A minor comment. In the conclusion (repeated in abstract and paper), I feel you could conclude beyond just this mouse model to provide some suggestion for human FM patients. As written the two concluding sentences also seem pretty much the same - about being careful not to induce arrhythmias in the mice. I would suggest a rewrite like following:

"Conclusion: Although a short-term free gait in shallow warm water may be advantageous for increasing the motor activity of FM-model mice, our data indicates it can be associated with changes in heart rate and even arrhythmias. Physiotherapists and other health professionals should be aware of these potential risks when considering strenuous exercise as a treatment in FM patients. We suggest a gradual increase in exercise duration may be warranted"

7. PLOS authors have the option to publish the peer review history of their article (what does this mean?). If published, this will include your full peer review and any attached files.

Reviewer #1: No

Reviewer #2: No

---

## [Editor Report · Acceptance letter]

15 Sep 2020

PONE-D-20-16502R1 

Excessive exercise induces cardiac arrhythmia in a young fibromyalgia mouse model 

Dear Dr. Doi:

I'm pleased to inform you that your manuscript has been deemed suitable for publication in PLOS ONE. Congratulations! Your manuscript is now with our production department. 

Kind regards, 

on behalf of

Prof. Etsuro Ito 

Academic Editor

PLOS ONE